# Effectiveness of erectogenic condom against semen exposure among women in Vietnam: Randomized controlled trial

Nghia C. Nguyen[1], Truong N. Luong[2], Van T. Le[2], Marcia Hobbs[3], Rebecca Andridge[4], John Casterline[5], Maria F. Gallo[6]*

1 Department of Obstetrics and Gynecology, Vinmec International Hospital, Hanoi, Vietnam, 2 Thanh Hoa Center for Disease Prevention and Control, Thanh Hoa City, Vietnam, 3 Department of Medicine, University of North Carolina at Chapel Hill, Chapel Hill, North Carolina, United States of America, 4 Division of Biostatistics, College of Public Health, Ohio State University, Columbus, Ohio, United States of America, 5 Department of Sociology, College of Arts and Sciences, Ohio State University, Columbus, Ohio, United States of America, 6 Division of Epidemiology, College of Public Health, Ohio State University, Columbus, Ohio, United States of America

* gallo.86@osu.edu

**Data Availability Statement:** The data and code for the primary paper are now at: https://osf.io/8d3b9/.

## Abstract

A key barrier to the consistent use of condoms is their negative effect on sexual pleasure. Although sexual pleasure is a primary motivation for engaging in sex and is an integral part of overall sexual health, most programs to improve sexual health operate within a pregnancy and disease-prevention paradigm. A new condom, CSD500 (Futura Medical Developments; Surrey, UK), containing an erectogenic drug was developed for use among healthy couples to improve sexual pleasure by increasing penile firmness, size and erection duration. We conducted a randomized controlled trial to test whether promoting the novel condom CSD500 for improved sexual pleasure is effective in reducing condomless sex compared to the provision of standard condoms with counseling for pregnancy and disease prevention. We randomized 500 adult, heterosexual, monogamous couples in Thanh Hoa province, Vietnam to receive either CSD500 (n = 248) or standard condoms (n = 252). At enrollment and after 2, 4, and 6 months, we interviewed women and sampled vaginal fluid to test for the presence of prostate-specific antigen (PSA), an objective, biological marker of recent semen exposure. We registered the protocol before trial initiation at ClinicalTrials.gov (identifier: NCT02934620). Overall, 11.0% of women were PSA positive at enrollment. The proportion of follow-up visits with PSA-positivity did not differ between the intervention (6.8%) and control arms (6.7%; relative risk, 1.01; 95% confidence interval, 0.66–1.54). Thus, we found no evidence that promoting an erectogenic condom to women in a monogamous, heterosexual relationship in Vietnam reduced their exposure to their partner's semen. These findings might not hold for other populations, especially those with a higher frequency of condomless sex.

**Funding:** Research reported in this publication was supported by the Eunice Kennedy Shriver National Institute of Child Health & Human Development of the National Institutes of Health under Award Number R01HD084637 and the National Center for Advancing Translational Sciences (UL1TR001070). The content is solely the responsibility of the authors and does not necessarily represent the official views of the funders. Futura Medical donated the CSD500 condoms used in the trial; they otherwise had no input into the study design or its report.

**Competing interests:** The authors have declared that no competing interests exist.

## Introduction

Despite advancements in biomedical interventions for preventing HIV and other sexually transmitted infections (STIs), condoms remain relevant. Male circumcision does not appear to lower women's HIV risk, and pre-exposure prophylaxis or treating those with HIV-infection to reduce infectivity have logistical challenges, including maintaining adherence to drug regimens or early identification of HIV infection [1, 2]. Furthermore, these interventions generally do not protect against other STIs. Condoms also are important for pregnancy prevention, especially in settings where concerns about the safety of hormonal contraception limit its use or among those wanting a coitally-dependent method due to, for example, infrequent sexual activity [3].

More than 80% of HIV cases in women worldwide result from sexual exposure to husbands or primary heterosexual partners [4]. Addressing condom use within established partnerships is a major public health challenge. A key barrier to adoption and consistent use of condoms is men's aversion to a product that interferes with sexual pleasure [5–8]. Male condoms are widely perceived to reduce pleasure by inhibiting spontaneity and restricting physical sensation. Impaired sensitivity can cause negative physical effects, including erection loss or inability to ejaculate [9–11]. An estimated 9%-37% of condoms users have had erection problems during condom application or use [9–11]. This experience–especially during initial, formative sexual encounters–can reduce men's confidence about their ability to maintain an erection during condom use, which may then cause a negative feedback loop of erectile dysfunction [10] and dissuade men from future attempts to use a condom. Even among those willing to use condoms, many report risky practices (e.g., delays in applying it or removal before ejaculation) in an effort to minimize the loss of pleasure [12–15]. At best, couples are neutral about the feel of condoms [8]; certainly condoms are not generally viewed as enhancing pleasure.

Pleasure is a primary motivator for sexual behavior [6, 8, 16] and is recognized by the World Health Organization and others as an integral part of sexual health [17]. Sexual health interventions, though, typically operate within a pregnancy and disease-prevention paradigm [18]. This failure to address sexuality, intimacy, eroticism, and pleasure may limit the successful promotion of condoms. Recently, a new condom CSD500 (Futura Medical Developments; Surrey, UK), that contains gel with 1% glyceryl trinitrate in the condom teat, was developed for use by healthy males to improve sexual pleasure by increasing penile firmness, size, and erection duration. We hypothesized that providing women in an established relationship with CSD500 accompanied by counseling focusing on the new condom's ability to improve male performance and pleasure would result in less condomless vaginal sex relative to providing a standard condom with traditional counseling focusing on disease and pregnancy prevention only. We used detection of prostate-specific antigen (PSA) in vaginal fluid as an objective measure of women's recent exposure to semen from penile-vaginal sex [19]. A future report will focus on the effects of CSD500 on male and female participants' sexual pleasure and condom acceptability.

## Materials and methods

### Study population

We enrolled 500 heterosexual couples at a large public health facility in Thanh Hoa, Vietnam during June 2017 to August 2019. To be eligible for participation, women needed to meet the following criteria: be 18–45 years of age, speak Vietnamese, not currently using modern contraception other than condoms, not intending to use a modern contraceptive method other than condoms in the next six months, and be in a monogamous relationship for at least the

past six months with her current male partner. Breastfeeding, known pregnancy and wanting a pregnancy in the next six months were exclusion criteria. Couples were ineligible for participation if either person was known to be HIV-positive or had a contraindication to CSD500 use (i.e., history of low blood pressure or heart condition; current use of medication for anemia, blood pressure, erectile dysfunction (man only), migraines, headaches, or glaucoma; inflamed or broken skin that the condom could come into contact with (man only); or latex allergy or sensitivity). Both the woman and her male partner must have been willing and able to provide written consent to study procedures, including the use of the assigned study condoms. Institutional review boards at the Ohio State University and the Hanoi School of Public Health approved the study (S1 and S2 Files). We registered the protocol before trial initiation at ClinicalTrials.gov (identifier: NCT02934620).

## Study procedures

Study interviewers recruited women attending the study site; if their male partner was not present, they were asked to return with him to complete the enrollment visit. Eligible couples who provided written consent were enrolled and randomized, using block randomization in REDCap, which served to conceal the allocation process, to one of two arms: 1) CSD500 for sexual pleasure or 2) the standard condom currently provided for pregnancy and disease prevention during routine care. Interviewers administered an enrollment questionnaire to men and women, separately, and recorded responses directly into REDCap [20]. A study clinician collected a double-headed vaginal swab per established procedures [21]. Study staff provided the assigned condom counseling to the couple together and distributed the assigned study condoms (20 condoms or more if needed based on expected coital frequency). Female participants were asked to return for follow up at 2, 4 and 6 months after enrollment, without adjusting for their menstrual cycle, to complete a follow-up questionnaire and to have another double-headed vaginal swab collected. The women's follow-up questionnaire asked about whether the woman or her partner (separate questions) had experienced any symptoms from wearing a study condom. Response options included those listed on the package insert for CSD500 (i.e., headaches, faintness, nausea, loss of sensation, dizziness, skin irritation) and a space to indicate and describe "other" side effects. At the 2 and 4-month visits only, women also received condom counseling and a resupply of condoms per their assigned arm. Male participants were asked to return at the 6-month visit to complete a follow-up questionnaire. All study materials were translated into Vietnamese and the consent form, questionnaires, and CSD500 package insert were back translated into English to ensure linguistic equivalence between the two versions. Study questionnaires were piloted with 10 couples from the target population. Although masking participants and clinic study staff to arm assignment was not feasible, the principal investigator and laboratory staff remained masked until the primary analyses were completed.

## Condom counseling

Staff provided standardized condom counseling on proper use of the assigned condom to couples at enrollment and to the female participants at the 2- and 4-month visits. The control arm received standard counseling on condom use for pregnancy and disease prevention without receiving any messages about condom use for sexual pleasure. The intervention arm received counseling that briefly addressed condom's dual protection against pregnancy and HIV/STI but that otherwise emphasized the potential for increased sexual pleasure with CSD500 use. The intervention arm also received the CSD500 package insert and CSD500-specific instructions, including the need to briefly massage the gel inside the condom teat onto the penis head

after donning the condom and not to use multiple condoms within a 24-hour period. CSD500 was referred to as "Futura Max" to participants.

## Prostate-Specific Antigen (PSA)

At each of the four scheduled study visits, a study clinician collected a vaginal fluid specimen, using a 1-mL, rayon-tipped double-headed swab, from female participants. Swabs were stored onsite at −70˚C following collection until their shipment in batches on dry ice to the laboratory at the University of North Carolina, where trained personnel processed and tested one of the double-headed swabs in batches for total PSA (Architect Total PSA; Abbott Diagnostics, Abbott Park, IL) following established procedures 21]. Detection of PSA in vaginal fluid is a marker of women's exposure to semen within the past 48 hours [19]. Because it is expressed independently of spermatozoa, PSA is useful for identifying exposure even from men who are vasectomized or otherwise without high levels of spermatozoa. False positive tests resulting from women's endogenous sources of PSA (e.g., serum or urine) are improbable given that the resulting PSA levels would be several orders of magnitude below the established threshold for defining PSA positivity from semen exposure. Because PSA begins to clear from vaginal fluid immediately after women's exposure to her partner's semen and is almost always undetectable by 48 hours post-exposure [19], the PSA outcome only measured semen exposure during the short interval preceding each of the three follow-up visits (rather than all semen exposure during follow up). Sampling a small portion of exposure is common, though, in behavioral research when continuous monitoring is infeasible; for example, condom use measures typically ask about use at last sex act as a way of reducing the potential for recall bias.

Preliminary data from an in vitro experiment conducted at the study laboratory in 2014 indicated that CSD500 condom does not interfere with PSA detection. Laboratory technicians spiked condom extracts (including material from the inside only, outside only, or a combination of both) and non-condom controls consisting of phosphate buffered saline (PBS) only with three concentrations of PSA (0.25, 2.00 and 11.50 ng/mL) and no PSA (control). Five independent samples of each condom extract condition were prepared for a total of 60 specimens and the PBS controls. Table 1 shows the mean

PSA detected for each condition, which were all within manufacturer ranges.

## Study outcomes

The primary outcome was the relative risk (RR) comparing PSA positivity (per established threshold of >1 ng of PSA per mL vaginal swab eluate) in the pooled 2, 4 and 6-month visits between study arms. Secondary analyses included 1) calculating the RR of PSA positivity at least once during follow up (i.e., using women, instead of visits, as the unit of analysis); 2)

**Table 1. Detection of PSA in PBS alone or material extracted from the inside or outside of CSD500 condoms from spiked specimens, by PSA concentrations used for spiking.**

| Spiked PSA concentration (ng/mL) | Architect Total PSA result (mean +/- standard deviation in ng/mL) for specimens prepared with [a] | | | |
|:---:|:---:|:---:|:---:|:---:|
| | **PBS** | **Condom inside** | **Condom outside** | **Condom in/out** |
| 0.00 | 0.00 +/- 0.00 | 0.00 +/- 0.00 | 0.00 +/- 0.00 | 0.00 +/- 0.00 |
| 0.25 | 0.23 +/- 0.01 | 0.23 +/- 0.01 | 0.22 +/- 0.01 | 0.22 +/- 0.01 |
| 2.00 | 1.88 +/- 0.03 | 1.88 +/- 0.08 | 1.89 +/- 0.07 | 1.95 +/- 0.12 |
| 11.50 | 11.74 +/- 0.29 | 11.76 +/- 0.59 | 11.71 +/- 0.59 | 12.08 +/- 0.31 |

[a] Each of the 16 test conditions (spiked PSA concentration by PBS or condom extract) evaluated for 5 independent specimens

PBS = phosphate buffered saline; PSA = prostate-specific antigen

using a higher threshold (4 ng/mL) to define PSA positivity; and 3) considering the effect of study duration on the main comparison.

### Sample size and statistical analysis

We estimated that a sample size of 500 subjects split equally in two arms would provide 80% power to detect a relative risk of PSA positivity of 0.61 between study arms with a two-tailed alpha of 0.05. We assumed based on previous studies [22–24], a baseline prevalence of positivity of 15% and a within-subject over-time correlation of r = 0.23. We also assumed that 10% of subjects would fail to return after enrollment and, among the 90% remaining in the study, the probability of missing a follow-up visit would be 25%.

All analyses were conducted using an intent-to-treat approach, and therefore included all randomized subjects [25]. Baseline characteristics of enrolled subjects were summarized by treatment arm using frequencies distributions. Analysis of primary and secondary outcomes were done using generalized estimating equations (GEE) with a log link and independent working correlation. All analyses were conducted using SAS version 9.4 (Cary, NC).

### Results

We screened 589 couples for eligibility, of whom 89 were excluded for either not meeting the inclusion criteria or declining to participate (Fig 1). We randomized the remaining 500 couples to the CSD500 intervention arm (n = 248) or the standard condom control arm (n = 252). The 2, 4 and 6-month follow-up visits were completed by 91.9%-94.4% of women in the intervention arm and 87.3%-94.8% of women in the control arm. Participants had a mean age of 33.9 years (standard deviation [SD], 5.3) and 34.3 years (SD, 5.5) in the intervention and control arms, respectively (Table 2). Almost all participants were of Kinh ethnicity and just over half (52% and 61% in the intervention and control arms, respectively) resided in a city. PSA-positivity was detected in 12% of women in the intervention arm and 10% in the control arm at enrollment.

The primary analysis did not find a difference in PSA-positivity at the follow-up visits between arms (Table 3). The proportion of follow-up visits with PSA detected was 6.8% (95% CI, 5.2%-8.9%) and 6.7% (95% CI, 4.8%-9.3%) in the intervention and control arms, respectively. The RR of PSA-positivity at follow-up visits in the intervention arm versus the control arm was 1.01 (95% CI, 0.66–1.54). Secondary analyses also detected no differences in arms; we repeated the primary analysis using a higher threshold for PSA-positivity (>4 ng/mL instead of >1 ng/mL) and found no difference between arms in the RR (0.80; 95% CI, 0.49–1.32). We also compared women with any PSA-positivity detected at a follow-up visit and found no different between the two arms (RR 1.25; 95% CI, 0.84–1.86). The overall proportion with PSA-positivity was lower during follow-up. In the intervention arm, PSA-positivity declined to 7.7% at the 2-month visit, 5.3% at the 4-month visit and 7.3% at the 6-month visit. Those in the control arm had similar decreases in the proportion with PSA-positivity: 8.8% at the 2-month visit, 5.5% at the 4-month visit and 5.9% at the 6-month visit.

During follow-up, more women in the CSD500 arm compared to the control arm (28% vs. 2.1%) reported experiencing at least one side effect related to the study condom (Table 4). Also, more women in the CSD500 arm compared to the control arm reported that their partner had experienced at least one side effect related to the study condom (15% vs. 1.3%). The most common side effects reported by women were headaches experienced by themselves (17% at 2-month, 14% at 4-month and 10% at the 6-month visit) and by their partners (6.9% at 2-month, 4.0% at 4-month and 4.3% at the 6-month visit).

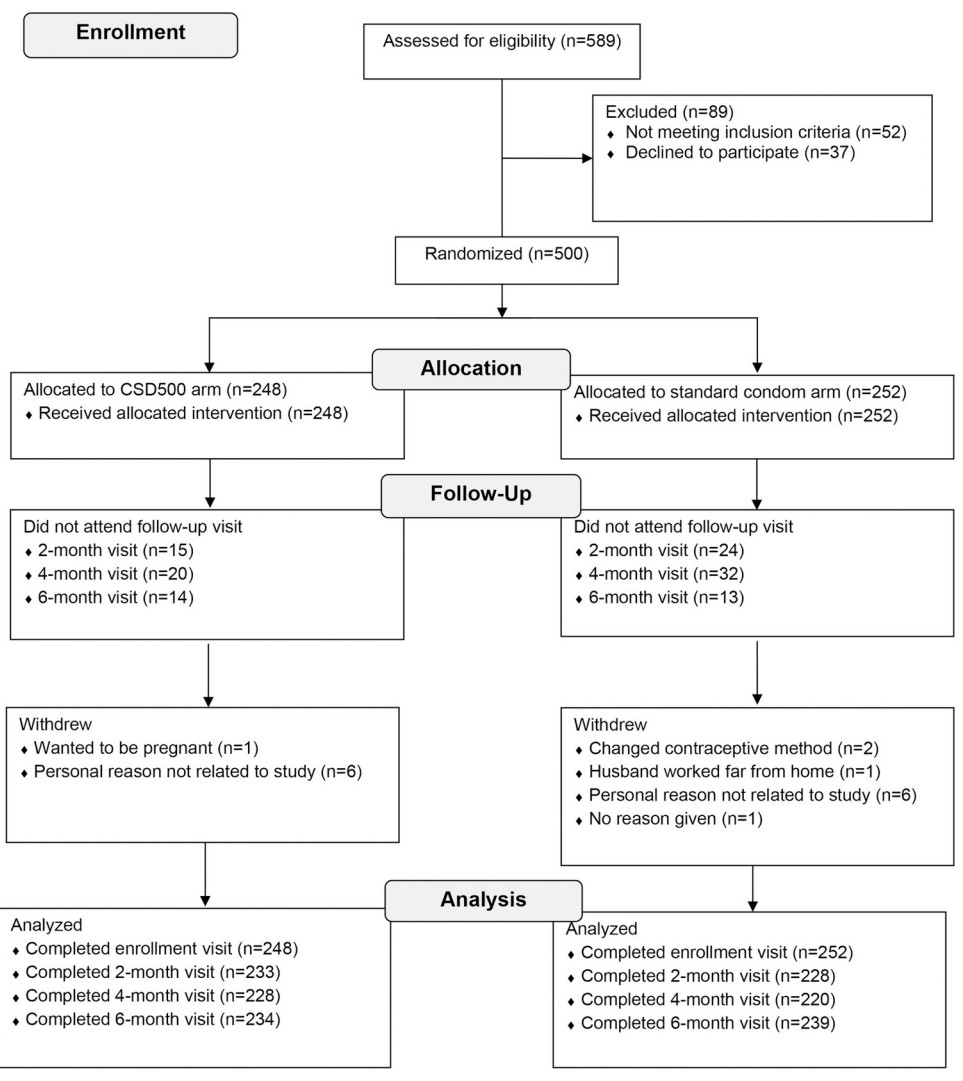

**Fig 1. Disposition of potential participants.**

## Discussion

Compared to a control group of women who received a standard condom and standard condom counseling, we did not find a decrease in PSA-positivity (i.e., the biomarker of women's recent exposure to semen) among those who were randomly assigned to use the erectogenic-containing CSD500 condom and were counseled that their male partner's use of the intervention condom could improve sexual pleasure by increasing his penile firmness, size, and erection duration. Despite widespread recognition of the role of condom use in protecting against pregnancy [26] and HIV/STI acquisition [27, 28], sustained use is notoriously difficult to achieve. Only 8.4% of reproductive-age women in the U.S. in 2017–2019 reported current condom use [29]. Similarly, a national survey in Vietnam found that only 9% and 7% of sexually-active, married, reproductive-age women reported condom use at last sex act and consistent condom use, respectively [30]. Furthermore, few studies have documented effectiveness of educational or counseling interventions in increasing condom use [21, 31, 32] and community-level, structural interventions to increase condom use have not been shown to decrease HIV/STI acquisition [33].

**Table 2. Characteristics of female participants at enrollment by study arm for intent-to-treat population (N = 500).**

|  | CSD500 arm (n = 248) | | Standard condom arm (n = 252) | |
|---|---|---|---|---|
|  | No. (%) or Mean (SD) | | No. (%) or Mean (SD) | |
| Age in years | 33.9 | (5.3) | 34.3 | (5.5) |
| Live births |  |  |  |  |
| 0 | 1 | (0.4%) | 1 | (0.4%) |
| 1 | 40 | (16.1%) | 32 | (12.7%) |
| 2 or more | 206 | (83.1%) | 217 | (86.1%) |
| Unknown | 1 | (0.4%) | 2 | (0.8%) |
| Ethnicity |  |  |  |  |
| Kinh | 245 | (98.8%) | 250 | (99.2%) |
| Non-Kinh | 3 | (1.2%) | 2 | (0.8%) |
| Highest level of education completed |  |  |  |  |
| Never attended school | 4 | (1.6%) | 3 | (1.2%) |
| Primary or lower secondary | 18 | (7.3%) | 18 | (7.1%) |
| Upper secondary | 49 | (20%) | 38 | (15%) |
| Higher | 177 | (71%) | 193 | (77%) |
| Residence |  |  |  |  |
| City | 130 | (52%) | 153 | (61%) |
| Town or rural area | 118 | (48%) | 99 | (39%) |
| PSA positivity |  |  |  |  |
| Yes | 29 | (12%) | 26 | (10%) |
| No | 219 | (88%) | 226 | (90%) |

Overall, 11.0% of women at enrollment were PSA positive compared to 6.7% of follow-up visits. That is, although the intervention condom did not result in less semen exposure during follow up relative to the control arm, the proportion with semen exposure among all study participants appeared to be lower during follow up. The provision of adequate supplies of condoms to participants could have caused a decline in condomless sex in both arms. This underscores the importance of including a control arm in evaluations of interventions; evidence from a single-arm, pre versus post-intervention study design might indicate that an intervention was effective in changing behavior when the changes instead were the result of participating in research. This finding also underscores the importance of providing condoms free of charge to the public. Women assigned to use CSD500 reported more side effects, in particular headaches, for both themselves and their partner. However, because this was an unblinded trial and because only women in the CSD500 arm were counseled on a list of possible side effects from their assigned condom, we cannot rule out that the higher frequency of side effects was the result of the nocebo effect (i.e., they perceived non-specific, negative side effects because they were primed to expect them to occur). Regardless, given that PSA-positivity decreased from enrollment to follow-up, the side effects of CSD500 did not appear to prevent their use.

The overall low proportion of PSA-positivity detected during follow-up visits suggests that the study population–sexually-active, non-contracepting women not desiring pregnancy who were in a monogamous, heterosexual relationship–were at low risk of semen exposure. Future evaluations of erectogenic condoms might be better suited for populations with a higher frequency of sex in general as well as condomless sex. Furthermore, the promotion of erectogenic condoms might have demonstrated effectiveness in a study population of more experienced condom users who were skilled at using the device. Finally, we provided the condom

**Table 3. PSA-positivity at follow-up by study arm for intent-to-treat population (N = 500).**

| | | CSD500 arm | | Standard condom arm | | | |
|---|---|---|---|---|---|---|---|
| | | No. | (%) | No. | (%) | RR | 95% CI |
| **Primary analysis** | | | | | | | |
| Follow-up visits with PSA-positivity[a] | | | | | | | |
| Yes | | 47 | (6.8%) | 46 | (6.7%) | 1.01 | (0.66, 1.54) |
| No | | 648 | (93.2%) | 641 | (93.3%) | | |
| **Secondary analyses** | | | | | | | |
| Follow-up visits with PSA-positivity[b] using higher threshold | | | | | | | |
| Yes | | 30 | (4.3%) | 37 | (5.4%) | 0.80 | (0.49, 1.32) |
| No | | 665 | (95.7%) | 650 | (94.6%) | | |
| Women with ≥1 follow-up visit with PSA-positivity[a] | | | | | | | |
| Yes | | 45 | (19%) | 36 | (15%) | 1.25 | (0.84, 1.86) |
| No | | 195 | (81%) | 204 | (85%) | | |
| Follow-up visits with PSA-positivity[a] | | | | | | | |
| 2-month visit | | | | | | | |
| Yes | | 18 | (7.7%) | 20 | (8.8%) | 0.88 | (0.48, 1.62) |
| No | | 215 | (92.3%) | 208 | (91.2%) | | |
| 4-month visit | | | | | | | |
| Yes | | 12 | (5.3%) | 12 | (5.5%) | 0.96 | (0.44, 2.10) |
| No | | 216 | (94.7%) | 208 | (94.5%) | | |
| 6-month visit | | | | | | | |
| Yes | | 17 | (7.3%) | 14 | (5.9%) | 1.24 | (0.63, 2.46) |
| No | | 217 | (92.7%) | 225 | (94.1%) | | |

CI = confidence interval; PSA = prostate-specific antigen; RR = relative risk

[a] Defined as >1 ng/mL

[b] Defined as >4 ng/mL

counseling to female participants (instead of men or couples) to more closely mimic non-study conditions, in which women typically bear the responsibility for condom and contraception uptake in clinic settings. Directly counseling men on the potential for improved sexual pleasure with CSD500 condoms, though, might be needed to achieve higher frequency of use.

CSD500 was designed to directly address condom-related erection loss. However, other negative properties of condoms might have prevented their consistent use in the present study. For example, condoms must be available and accessible during the sex act and the physical lack of spontaneity or interruption to the flow of sex could reduce sexual pleasure. Unappealing physical properties (related to touch, taste, or smell) of condoms could prevent their use. For example, poor fit of the condom, cold feel of latex, difficulty in donning the condom and doing so without losing an erection could limit the use of condoms [9–15]. Also, some might experience reduced emotional closeness from the act of wearing a condom or might interpret the inability to ejaculate into the vagina while wearing a condom as a reduction in their expression of male vitality [5, 10]. These negative condom attributes–inherent to all condom types, including the intervention condom–might prove intractable for a subset of individuals.

Primary strengths of the study included its randomized design and use of a biomarker of semen exposure instead of relying on self-reported use of condoms. Past evaluations of

**Table 4. Side effects reported by female participants during the follow-up period by study arm.**

| | | | CSD500 arm | | Standard condom arm | | |
|---|---|---|---|---|---|---|---|
| Woman reported any side effect during follow-up | | | | | | | |
| | For herself | | 28% | (68/240) | 2.1% | (5/240) | |
| | For her partner | | 15% | (36/240) | 1.3% | (3/240) | |
| Specific side effects for the woman | | | | | | | |
| | Headaches | 2-month | 17% | (40/233) | 0% | (0/227) | |
| | | 4-month | 14% | (32/227) | 0% | (0/220) | |
| | | 6-month | 10% | (24/233) | 0% | (0/238) | |
| | Faintness | 2-month | 0.9% | (2/233) | 0% | (0/227) | |
| | | 4-month | 0.9% | (2/227) | 0% | (0/220) | |
| | | 6-month | 0% | (0/233) | 0% | (0/238) | |
| | Nausea | 2-month | 1.7% | (4/233) | 0% | (0/227) | |
| | | 4-month | 1.8% | (4/227) | 0% | (0/220) | |
| | | 6-month | 0% | (1/233) | 0% | (0/238) | |
| | Loss of sensation | 2-month | 1.3% | (3/233) | 0.4% | (1/227) | |
| | | 4-month | 0% | (0/227) | 0% | (0/220) | |
| | | 6-month | 0% | (0/233) | 0% | (0/238) | |
| | Dizziness | 2-month | 10% | (24/233) | 0.4% | (1/227) | |
| | | 4-month | 5.3% | (12/227) | 0.5% | (1/220) | |
| | | 6-month | 4.3% | (10/233) | 0% | (0/238) | |
| | Skin irritation | 2-month | 1.3% | (3/233) | 0% | (0/227) | |
| | | 4-month | 0.9% | (2/227) | 0% | (0/220) | |
| | | 6-month | 0.4% | (1/233) | 0% | (0/238) | |
| Specific side effects for the man | | | | | | | |
| | Headaches | 2-month | 6.9% | (16/233) | 0% | (0/227) | |
| | | 4-month | 4.0% | (9/227) | 0% | (0/220) | |
| | | 6-month | 4.3% | (10/233) | 0% | (0/238) | |
| | Faintness | 2-month | 1.7% | (4/233) | 0% | (0/227) | |
| | | 4-month | 0.4% | (1/227) | 0% | (0/220) | |
| | | 6-month | 0% | (0/233) | 0% | (0/238) | |
| | Nausea | 2-month | 0% | (0/233) | 0% | (0/227) | |
| | | 4-month | 0% | (0/227) | 0% | (0/220) | |
| | | 6-month | 0% | (0/233) | 0% | (0/238) | |
| | Loss of sensation | 2-month | 0.4% | (1/233) | 0.4% | (1/227) | |
| | | 4-month | 0% | (0/227) | 0% | (0/220) | |
| | | 6-month | 0% | (1/233) | 0% | (0/238) | |
| | Dizziness | 2-month | 4.3% | (10/233) | 0% | (0/227) | |
| | | 4-month | 1.3% | (3/227) | 0% | (0/220) | |
| | | 6-month | 2.1% | (5/233) | 0% | (0/238) | |
| | Skin irritation | 2-month | 0.9% | (2/233) | 0% | (0/227) | |
| | | 4-month | 0% | (0/227) | 0% | (0/220) | |
| | | 6-month | 0% | (0/233) | 0% | (0/238) | |

condom promotion interventions have largely relied on participant reports of risky sexual behavior, which can have poor validity for many reasons: social desirability or recall bias, lack of awareness of exposure (e.g., undetected condom breakage), or poor comprehension of or embarrassment about the survey questions [34]. Studies using semen biomarkers to validate participant reports show that under-reporting of semen exposure is common [19] and can

vary by study and participant factors, including HIV risk [35]. Thus, findings from previous condom promotion studies that relied on participant reports could be biased and lead to incorrect conclusions. Whether the study findings are generalizable to a non-research setting or other populations remains unknown.

In summary, we found no evidence that promoting an erectogenic condom to women results in their having less exposure to their partner's semen. These findings might not hold for other populations. Relatively few women in the present study had PSA detected at enrollment, indicating infrequent exposure to semen from condomless sex and that the possible ceiling for finding an effect from the intervention condom was low. Also, PSA-positivity decreased in both study arms during follow up. Participants might have been adherent to the clinician's instructions and the study counseling messages to use their assigned condoms consistently for sex during the study. Future research could test the CSD500 condom in populations that engage in more condomless sex and that might be less adherent to study messages.

## Supporting information

**S1 Checklist. CONSORT 2010 checklist of information to include when reporting a randomised trial**[*].
(DOC)

**S1 File. Study protocol.**
(PDF)

**S2 File. Study questionnaires.**
(DOC)

## Author Contributions

**Conceptualization:** Nghia C. Nguyen, Marcia Hobbs, Rebecca Andridge, John Casterline, Maria F. Gallo.

**Data curation:** Nghia C. Nguyen, Truong N. Luong, Van T. Le, Marcia Hobbs, Rebecca Andridge, Maria F. Gallo.

**Formal analysis:** Rebecca Andridge, Maria F. Gallo.

**Funding acquisition:** Nghia C. Nguyen, Marcia Hobbs, Rebecca Andridge, John Casterline, Maria F. Gallo.

**Investigation:** Nghia C. Nguyen, Truong N. Luong, Van T. Le, Marcia Hobbs, Rebecca Andridge, Maria F. Gallo.

**Methodology:** Nghia C. Nguyen, Van T. Le, Marcia Hobbs, Rebecca Andridge, John Casterline, Maria F. Gallo.

**Project administration:** Nghia C. Nguyen, Truong N. Luong, Van T. Le, Marcia Hobbs, Maria F. Gallo.

**Resources:** Maria F. Gallo.

**Software:** Rebecca Andridge, Maria F. Gallo.

**Supervision:** Nghia C. Nguyen, Truong N. Luong, Van T. Le, Marcia Hobbs.

**Validation:** Nghia C. Nguyen, Truong N. Luong, Van T. Le, Marcia Hobbs, Rebecca Andridge, Maria F. Gallo.

**Visualization:** Maria F. Gallo.

**Writing – original draft:** Maria F. Gallo.

**Writing – review & editing:** Nghia C. Nguyen, Truong N. Luong, Van T. Le, Marcia Hobbs, Rebecca Andridge, John Casterline, Maria F. Gallo.

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
