## [Decision Letter · Decision Letter 0]

23 Oct 2021

PONE-D-21-12418Effectiveness of erectogenic condom against semen exposure among women in Vietnam: Randomized controlled trial PLOS ONE

Dear Dr. Gallo,

Thank you for submitting your manuscript to PLOS ONE. After careful consideration, we feel that it has merit but does not fully meet PLOS ONE’s publication criteria as it currently stands. Therefore, we invite you to submit a revised version of the manuscript that addresses the points raised during the review process.

The manuscript has been evaluated by three reviewers, and their comments are available below. The reviewers have raised a number of concerns that need attention. They request additional information on methodological aspects of the study, interpretation of the results, as well as more information regarding the data supporting the effect of erectogenic drugs. Could you please revise the manuscript to carefully address the concerns raised?

We look forward to receiving your revised manuscript.

Kind regards,

Dario Ummarino, Ph.D.

Senior Editor

PLOS ONE

Journal Requirements:

2. Thank you for including your ethics statement:  "IRBs of the Ohio State University and Hanoi School of Public Health. 2015H0242 Written consent".  

Please amend your current ethics statement to confirm that your named institutional review board or ethics committee specifically approved this study.

3. Please include additional information regarding the survey or questionnaire used in the study and ensure that you have provided sufficient details that others could replicate the analyses. For instance, if you developed a questionnaire as part of this study and it is not under a copyright more restrictive than CC-BY, please include a copy, in both the original language and English, as Supporting Information. If the original language is written in non-Latin characters, for example Amharic, Chinese, or Korean, please use a file format that ensures these characters are visible."

4. Please state whether you validated the questionnaire prior to testing on study participants. Please provide details regarding the validation group within the methods section.

Reviewers' comments:

Reviewer's Responses to Questions

**Comments to the Author**

1. Is the manuscript technically sound, and do the data support the conclusions?

Reviewer #1: Yes

Reviewer #2: Yes

Reviewer #3: Yes

2. Has the statistical analysis been performed appropriately and rigorously? 

Reviewer #1: Yes

Reviewer #2: Yes

Reviewer #3: Yes

3. Have the authors made all data underlying the findings in their manuscript fully available?

Reviewer #1: Yes

Reviewer #2: Yes

Reviewer #3: Yes

4. Is the manuscript presented in an intelligible fashion and written in standard English?

Reviewer #1: Yes

Reviewer #2: Yes

Reviewer #3: Yes

5. Review Comments to the Author

Reviewer #1: A two-arm randomized controlled clinical trial was conducted which aimed to compare the proportion of Vietnamese couples using a novel condom to controls, as measured by the presence of PSA in vaginal fluid. The proportion of those PSA positive did not significantly differ in the two arms. The manuscript is thorough and exceptionally well written.

Minor revisions:

1- Line 181: Perhaps the word “age” has accidentally been omitted.

2- Line 193: Provide 95% confidence intervals (CI) for the proportions where PSA was detected, i.e. CIs for 6.8% and 6.7%.

Reviewer #2: The authors tried to find out whether promoting a new type of condom, recently developed for use by healthy males to improve sexual pleasure increasing penile firmness, size and erection duration, could reduce exposure of women to the semen of their partners preventing HIV and other sexually transmitted infections.

The study is well done and well presented.

Reviewer #3: This is a well-written paper with a negative finding that I believe will deserve publication, if for no other reason, than to validate the use of biochemical markers in condom studies in lieu traditional efficacy trials. However, there are a few general and specific comments I would appreciate the authors to address.

General comments

1. The claim is made that some drug added to the condom would improve different aspects of erection, but no data are provided to support these claims, and no validation was obtained from the subjects that such changes had occurred. That would be a critical element to learn. Perhaps couples were disappointed by the minor differences they experienced. Perhaps an agent that did not require extra steps in application could still affect increased utilization. The authors should elaborate more on these points.

2. Along those lines, apparently no information was collected about potential adverse side effects that could have influenced use patterns.

3. Were there any efforts to measure subject satisfaction? Discontinuation rates were low in both arms.

4. The authors noted that frequency of detectable PSA declined from baseline but did not address the impact that providing subjects with condoms could have had on that decline. This is important since the decline happened in both groups.

5. Similarly, were the subjects asked at each visit if they had adequate supplies? Could the PSA rise reflect lack of available condoms? I suspect they had been told not to use other condoms.

6. Another point that should be discussed is what percent of subjects in each arm were previously successful condom users. There are clear differences in use patterns between new users and experienced users. That is seen in all clinical trials with new condom designs. (See below).

7. Other problems that latex condom users face should also be mentioned:

a. Size problems (up to 30% of men can not fit in existing sizes). This issue gets back to point 5 – unless we know the man can use the condom, then the impact of the compound is not being reflected. This might be important to interpret results if the authors found a cohort of women who consistently tested positive.

b. Temperature problems. Latex is cold and can contribute to loss of erection during placement and discomfort for the woman.

c. Difficulty unrolling (placing) the condom. Again, the use of inexperienced couples could have masked the impact of the drug.

Were any of these problems addressed in the condoms that were used in the study? If not, perhaps they should be mentioned. Alternatively, the authors could report the outcomes of experienced condom-using couples to see if the compound did influence use or perhaps that should be the design of the next study.

Small points

Line 47-48: Could delete “in setting. . .”. Condoms are important for pregnancy protection.

Line 51: Perhaps you could add “heterosexual” to partnerships to highlight focus on this study.

Line 54: Could add the extra condom-associated problems listed above.

Line 107: What are results of the male questionnaire?

Line 122: Should probably include anticipated multiple users as an exclusionary criterion.

Line 141: Why did you not also ask about recent coitus to validate findings?

Line 189: If you had coital history, you could calculate more meaningful estimates - % of subjects who had sex and positive PSA.

Line 225 – Also underscores importance of providing condoms for free.

Line 233: Very good point.

Line 246: Please list other condom-use problems. Some of those may be solvable.

I believe this paper has great potential and look forward to reviewing any revisions the authors choose to submit.

6. PLOS authors have the option to publish the peer review history of their article (what does this mean?). If published, this will include your full peer review and any attached files.

Reviewer #1: No

Reviewer #2: No

Reviewer #3: No

---

## [Author Response · Author response to Decision Letter 0]

12 Nov 2021

November 8, 2021

Dario Ummarino, Ph.D.

Senior Editor

PLOS ONE

RE: Resubmission of Manuscript PONE-D-21-12418

Dear Dr. Ummarino,

Thank you for the opportunity to respond to the second round of review of our manuscript, PONE-D-21-12418 “Effectiveness of erectogenic condom against semen exposure among women in Vietnam: Randomized controlled trial.” Our responses to the Journal Requirements and reviewers are as follows:

Journal Requirements:

We confirm that we followed these instructions.

2. Thank you for including your ethics statement: "IRBs of the Ohio State University and Hanoi School of Public Health. 2015H0242 Written consent". Please amend your current ethics statement to confirm that your named institutional review board or ethics committee specifically approved this study. Once you have amended this/these statement(s) in the Methods section of the manuscript, please add the same text to the “Ethics Statement” field of the submission form (via “Edit Submission”). For additional information about PLOS ONE ethical requirements for human subjects research, please refer to http://journals.plos.org/plosone/s/submission-guidelines#loc-human-subjects-research.

We amended the ethics statement in lines 93-94 to read: “Institutional review boards at the Ohio State University and the Hanoi School of Public Health approved the study.” We added the same text to the online submission system.

3. Please include additional information regarding the survey or questionnaire used in the study and ensure that you have provided sufficient details that others could replicate the analyses. For instance, if you developed a questionnaire as part of this study and it is not under a copyright more restrictive than CC-BY, please include a copy, in both the original language and English, as Supporting Information. If the original language is written in non-Latin characters, for example Amharic, Chinese, or Korean, please use a file format that ensures these characters are visible."

We are including the English and Vietnamese versions of the questionnaires as Supporting Information with this submission.

4. Please state whether you validated the questionnaire prior to testing on study participants. Please provide details regarding the validation group within the methods section.

We added new text in lines 116-117 on the process of questionnaire development: 

“All study materials were translated into Vietnamese and the consent form, questionnaires, and CSD500 package insert were back translated into English to ensure linguistic equivalence between the two versions. Study questionnaires were piloted with 10 couples from the target population.”

Not applicable to this submission.

We reviewed and have no changes.

Reviewer: 1

A two-arm randomized controlled clinical trial was conducted which aimed to compare the proportion of Vietnamese couples using a novel condom to controls, as measured by the presence of PSA in vaginal fluid. The proportion of those PSA positive did not significantly differ in the two arms. The manuscript is thorough and exceptionally well written.

1. Line 181: Perhaps the word “age” has accidentally been omitted.

Thank you for noticing this. We added the word “age” in line 188.

2. Line 193: Provide 95% confidence intervals (CI) for the proportions where PSA was detected, i.e. CIs for 6.8% and 6.7%.

We added the following text (new text is underlined) to lines 201-202: The proportion of follow-up visits with PSA detected was 6.8% (95% CI, 5.2%-8.9%) and 6.7% (95% CI, 4.8%-9.3%) in the intervention and control arms, respectively.

Reviewer #2: 

The authors tried to find out whether promoting a new type of condom, recently developed for use by healthy males to improve sexual pleasure increasing penile firmness, size and erection duration, could reduce exposure of women to the semen of their partners preventing HIV and other sexually transmitted infections.

The study is well done and well presented.

Reviewer #3: 

1. The claim is made that some drug added to the condom would improve different aspects of erection, but no data are provided to support these claims, and no validation was obtained from the subjects that such changes had occurred. That would be a critical element to learn. Perhaps couples were disappointed by the minor differences they experienced. Perhaps an agent that did not require extra steps in application could still affect increased utilization. The authors should elaborate more on these points.

We have a paper under review that consists of a detailed analysis on the effects of the condom on men and women’s sexual pleasure and acceptability of their assigned condom. This analysis found that those assigned to use CSD500 (both men and women) reported greater sexual pleasure and condom acceptability relative to those in the control arm. Unfortunately, though, this cannot yet be cited. We added the following sentence to lines 76-77 to explain this: “A future report will focus on the effects of CSD500 on male and female participants’ sexual pleasure and condom acceptability.”

2. Along those lines, apparently no information was collected about potential adverse side effects that could have influenced use patterns.

We collected data on side effects and added this to this manuscript. Specifically, we added the following text to lines 109-112 in the Methods section: “The women’s follow-up questionnaire asked about whether the woman or her partner (separate questions) had experienced any symptoms from wearing a study condom. Response options included those listed on the package insert for CSD500 (i.e., headaches, faintness, nausea, loss of sensation, dizziness, skin irritation) and a space to indicate and describe “other” side effects.”

These results are reported in the Results section in lines 217-223: “During follow-up, more women in the CSD500 arm compared to the control arm (28% vs. 2.1%) reported experiencing at least one side effect related to the study condom (Table 4). Also, more women in the CSD500 arm compared to the control arm reported that their partner had experienced at least one side effect related to the study condom (15% vs. 1.3%). The most common side effects reported by women were headaches experienced by themselves (17% at 2-month, 14% at 4-month and 10% at the 6-month visit) and by their partners (6.9% at 2-month, 4.0% at 4-month and 4.3% at the 6-month visit).”

We also added text in the Discussion section in lines 249-255: “Women assigned to use CSD500 reported more side effects, in particular headaches, for both themselves and their partner. However, because this was an unblinded trial and because only women in the CSD500 arm were counseled on a list of possible side effects from their assigned condom, we cannot rule out that the higher frequency of side effects was the result of the nocebo effect (i.e., they perceived non-specific, negative side effects because they were primed to expect them to occur). Regardless, given that PSA-positivity decreased from enrollment to follow-up, the side effects of CSD500 did not appear to prevent their use.”

3. Were there any efforts to measure subject satisfaction? Discontinuation rates were low in both arms.

We used 11 items to measure condom acceptability; an analysis of these items found greater acceptability in the CSD500 arm compared to the control arm. These findings are currently under review, and unfortunately cannot be cited yet. We added the following sentence to lines 76-77 to explain this: “A future report will focus on the effects of CSD500 on male and female participants’ sexual pleasure and condom acceptability.”

4. The authors noted that frequency of detectable PSA declined from baseline but did not address the impact that providing subjects with condoms could have had on that decline. This is important since the decline happened in both groups.

This is an important point. We added the following text to the Discussion section (lines 244-245) where we discuss the decrease in PSA-positivity from enrollment to the follow-up visits: “The provision of adequate supplies of condoms to participants could have caused a decline in condomless sex in both arms.”

5. Similarly, were the subjects asked at each visit if they had adequate supplies? Could the PSA rise reflect lack of available condoms? I suspect they had been told not to use other condoms.

Participants in both arms were provided with more condoms than that anticipated to be needed based on the individual couple’s coital frequency (see text in line 104). Given the study procedures related to dispensing adequate condom supplies at the study visits and given the lack of reports from participants regarding lacking condoms, it seems unlikely that the condom supplies were inadequate.

6. Another point that should be discussed is what percent of subjects in each arm were previously successful condom users. There are clear differences in use patterns between new users and experienced users. That is seen in all clinical trials with new condom designs. (See below).

a. Other problems that latex condom users face should also be mentioned:

Size problems (up to 30% of men can not fit in existing sizes). This issue gets back to point 5 – unless we know the man can use the condom, then the impact of the compound is not being reflected. This might be important to interpret results if the authors found a cohort of women who consistently tested positive.

b. Temperature problems. Latex is cold and can contribute to loss of erection during placement and discomfort for the woman.

Difficulty unrolling (placing) the condom. Again, the use of inexperienced couples could have masked the impact of the drug.

Were any of these problems addressed in the condoms that were used in the study? If not, perhaps they should be mentioned. Alternatively, the authors could report the outcomes of experienced condom-using couples to see if the compound did influence use or perhaps that should be the design of the next study.

Male participants were asked at enrollment about their frequency of condom use with their current partner and whether they had ever used a condom with any partner. Unfortunately, though, we did not collect data for classifying participants as 1) experienced condom users vs inexperienced or 2) successful condom users vs unsuccessful users. Given the randomized design, we can expect the proportions to be similar between arms. As the reviewer points out, though, we might have failed to detect effectiveness of the CSD500 condom if the study population consisted of predominately inexperienced or unsuccessful condom users. This seems unlikely as the level of PSA-positivity was relatively low. We added the following text to discuss the possibility that condom problems limited the use of condoms in the study (lines 259-261): “Furthermore, the promotion of erectogenic condoms might have demonstrated effectiveness in a study population of more experienced condom users who were skilled at using the device.9-15”

We added the following text (new text is underlined) to lines 271-273: “For example, poor fit of the condom, cold feel of latex, difficulty in donning the condom and doing so without losing an erection could limit the use of condoms.”

7. Line 47-48: Could delete “in setting. . .”. Condoms are important for pregnancy protection.

We included additional text (new text is underlined) to highlight more examples of when condoms can be particularly important for pregnancy protection: “Condoms also are important for pregnancy prevention, especially in settings where concerns about the safety of hormonal contraception limit its use or among those wanting a coitally-dependent method due to, for example, infrequent sexual activity.”

8. Line 51: Perhaps you could add “heterosexual” to partnerships to highlight focus on this study.

We added the word “heterosexual” here.

9. Line 54: Could add the extra condom-associated problems listed above.

We added the following new text in lines 271-273: “For example, poor fit of the condom, cold feel of latex, difficulty in donning the condom and doing so without losing an erection could limit the use of condoms.”

10. Line 107: What are results of the male questionnaire?

The male and female questionnaires included measures of sexual pleasure and condom acceptability. A detailed report on these findings is under review.

11. Line 122: Should probably include anticipated multiple users as an exclusionary criterion.

We did not include this as an exclusion criterion for study participation and thus cannot restrict on this factor. Our intent was to try to keep the study population as general as possible to improve the generalizability of the study findings. Also, it is possible that past history of needing to try multiple condoms in a single act wouldn’t predict this occurrence with the intervention condom.

12. Line 141: Why did you not also ask about recent coitus to validate findings?

We asked about recent coitus and condom use, and future exploratory analyses could assess the agreement between PSA-positivity and self-reported exposure to semen from condomless sex. However, we don’t expect this to yield novel findings. A large body of literature – including studies using semen biomarkers as the referent standard – suggests that self-reports of sex and condom use have poor validity. For this reason, we decided a priori to use PSA-positivity as the outcome measure. 

13. Line 189: If you had coital history, you could calculate more meaningful estimates - % of subjects who had sex and positive PSA.

We decided a priori to use PSA-positivity as the outcome measure (and reported this to ClinicalTrials.gov) and decided against using a combined measure of self-reported recent condomless sex and/or PSA-positivity. This decision was based on research suggesting that self-reports are not only inaccurate but likely are biased, i.e., their accuracy can vary by study and participant factors, including HIV risk (e.g., see Gallo MF, Steiner MJ, Hobbs MM, Warner L, Jamieson DJ, Macaluso M. Biological markers of sexual activity: tools for improving measurement in HIV/STI prevention research. Sex Transm Dis 2013;40:447-52). This literature suggests that including self-reports of sex and condom use could itself introduce bias rather than improve the measure of the occurrence of condomless sex. We attempt to explain our rationale to not rely on self-reported data in lines 257-263.

14. Line 225 – Also underscores importance of providing condoms for free.

We added the following sentence to lines 248-249: “This finding also underscores the importance of providing condoms free of charge to the public.”

15. Line 233: Very good point.

Thank you.

16. Line 246: Please list other condom-use problems. Some of those may be solvable.

We expanded our discussion lines 259-261): “Furthermore, the promotion of erectogenic condoms might have demonstrated effectiveness in a study population of more experienced condom users who were skilled at using the device.9-15”

We hope the revisions are acceptable to the reviewers and appreciate the consideration of our submission for publication.

---

## [Decision Letter · Decision Letter 1]

21 Jan 2022

Effectiveness of erectogenic condom against semen exposure among women in Vietnam: Randomized controlled trial

PONE-D-21-12418R1

Dear Dr. Gallo,

We’re pleased to inform you that your manuscript has been judged scientifically suitable for publication and will be formally accepted for publication once it meets all outstanding technical requirements.

Kind regards,

James Mockridge

Staff Editor

PLOS ONE

Additional Editor Comments:

The manuscript has been assessed by three reviewers who find that the concerns about the methodological aspects of the study and interpretation of the results have been sufficiently addressed. Reviewer #3 suggests one minor detail in the Introduction regarding the design of the condom, which the authors may consider including in their final version for clarification

Reviewers' comments:

Reviewer's Responses to Questions

**Comments to the Author**

1. If the authors have adequately addressed your comments raised in a previous round of review and you feel that this manuscript is now acceptable for publication, you may indicate that here to bypass the “Comments to the Author” section, enter your conflict of interest statement in the “Confidential to Editor” section, and submit your "Accept" recommendation.

Reviewer #1: All comments have been addressed

Reviewer #2: All comments have been addressed

Reviewer #3: (No Response)

2. Is the manuscript technically sound, and do the data support the conclusions?

Reviewer #1: (No Response)

Reviewer #2: Yes

Reviewer #3: Yes

3. Has the statistical analysis been performed appropriately and rigorously? 

Reviewer #1: (No Response)

Reviewer #2: Yes

Reviewer #3: I Don't Know

4. Have the authors made all data underlying the findings in their manuscript fully available?

Reviewer #1: (No Response)

Reviewer #2: Yes

Reviewer #3: Yes

5. Is the manuscript presented in an intelligible fashion and written in standard English?

Reviewer #1: (No Response)

Reviewer #2: Yes

Reviewer #3: Yes

6. Review Comments to the Author

Reviewer #1: (No Response)

Reviewer #2: Corrections, made according to useful suggestions by the reviewer 3, improved the manuscript.

I have no any additional comments.

Reviewer #3: I thank the authors for their detailed responses to my earlier comments. I think that it is always important to publish negative results, but in this case, publications particularly important because this manuscript is very thought provoking. The only suggestions I would make to the authors would be that in the introduction they describe that the compound is packed in the tip of the condom and should be retained within the inside of the condom with use, so women should generally not be exposed to the drug. This is important to understand when evaluating the reported side effects. Even if the authors do not agree to this change, I would support publication of the manuscript. I do not need to see if again before it gets accepted.

7. PLOS authors have the option to publish the peer review history of their article (what does this mean?). If published, this will include your full peer review and any attached files.

Reviewer #1: No

Reviewer #2: No

Reviewer #3: No

---

## [Editor Report · Acceptance letter]

9 Feb 2022

PONE-D-21-12418R1 

Effectiveness of erectogenic condom against semen exposure among women in Vietnam:
Randomized controlled trial 

Dear Dr. Gallo:

I'm pleased to inform you that your manuscript has been deemed suitable for publication in PLOS ONE. Congratulations! Your manuscript is now with our production department. 

Kind regards, 

on behalf of

Dr James Mockridge 

Staff Editor

PLOS ONE